# The Cyprus Institute of Neurology and Genetics, an emerging paradigm of a gender egalitarian organisation

Stavroulla Xenophontos[1]*, Margarita Zachariou[2], Pavlos Polycarpou[1], Elena Ioannidou[3], Vera Kazandjian[3], Maria Lagou[3], Anna Michaelidou[3], George M. Spyrou[2], Marios A. Cariolou[1], Leonidas Phylactou[4]

1 Laboratory of Forensic Genetics & the Department of Cardiovascular Genetics, The Cyprus Institute of Neurology & Genetics, Nicosia, Cyprus, 2 Bioinformatics Department, The Cyprus Institute of Neurology & Genetics, Nicosia, Cyprus, 3 Finance and Administration Department, The Cyprus Institute of Neurology & Genetics, Nicosia, Cyprus, 4 Molecular Genetics, Function & Therapy Department, The Cyprus Institute of Neurology & Genetics, Nicosia, Cyprus

☯ These authors contributed equally to this work.
* sxenofon@cing.ac.cy

**Data Availability Statement:** All relevant data are within the manuscript and its Supporting Information files. For the raw data downloaded

## Abstract

Females are underrepresented in the science, technology, engineering, mathematics and medicine (STEMM) disciplines globally and although progress has been made, the gender gap persists. Our aim was to explore gender parity in the context of gender representation and internal collaboration at the Cyprus Institute of Neurology and Genetics (CING), a leading national biomedical organisation accredited as an equal opportunity employer. Towards this aim we (1) explored trends in gender parity within the different departments, positions and qualifications and in student representation in the CING's postgraduate school and, (2) investigated the degree of collaboration between male and female researchers within the Institute and the degree of influence within its co-authorship network. We recorded an over-representation of females both in the CING employees and the postgraduate students. The observed female over-representation in pooled CING employees was consistent with a similar over-representation in less senior positions and was contrasted with an observed male over-representation in only one middle rank and culminated in gender equality in the top rank in employee hierarchy. In terms of collaboration, both males and females tended to collaborate with each other without any significant preference to either inter-group or intra-group collaboration. Further comparison of the two groups with respect to their influence in the network in terms of occupying the positions of highest centrality scores, indicated that both gender and seniority level (head vs non-head) were significant in shaping the authors' influence, with no significant difference in those belonging in the same seniority level with respect to their gender. To conclude, our study has validated the formal recognition of the CING's policies and procedures pertinent to its egalitarian culture through the majority of the metrics of gender equality assessed in this study and has provided an extendable paradigm for evaluating gender parity in academic organizations.

from Scopus the website and search term is provided in the methods.

**Funding:** Margarita Zachariou and George M. Spyrou were funded by the European Commission Research Executive Agency (REA), Grant BIORISE (Num.669026), under the Spreading Excellence, Widening Participation, Science with and for Society Framework. The funders had no role in study design, data collection and analysis, decision to publish, or preparation of the manuscript.

**Competing interests:** The authors have declared that no competing interests exist.

## Introduction

Gender equality has been a perpetual issue of concern and debate from the beginning of human civilization as exemplified by the opposing views of the ancient Greek philosophers Aristotle and Plato, and, will most likely continue to be so for many years to come [1, 2]. Gender inequality and more specifically female gender discrimination in employment and from a social perspective, was formerly a widespread and deeply rooted manifestation in developed countries. It was not until the emergence of the second feminist movement in the 19th century throughout Europe and America that state policies and legal remedies to this social problem were embraced for the improvement of the quality of life of females [3–5]. However, irrespective of these legal remedies and social reforms, there are still many gender gaps as documented in the recent enlightening 2021 EU Report on the Gender Equality Index [6]. The latter report, describes in great detail both the achievements of gender equality policy initiatives to date, as well as the necessary improvements with respect to six core domains (*work*, *money*, *knowledge*, *time*, *power & health*). One can appreciate that these domains encapsulate fundamental themes evolving from the concepts of basic human rights in a modern society as laid down in the relevant EU law [6]. The 2021 EU Report on the Gender Equality Index uses a scale of 1 to 100, where 1 is for total inequality and 100 is for total equality. Cyprus's overall score is 57 and is ranked 21st in the EU. In the domains of knowledge and work, Cyprus's score is 56 and 70.6 respectively.

In this report, social implications were discussed and the need for improving the metrics of equality was placed into perspective. The persistent gender inequality in employment is succinctly addressed by the Council of Europe Gender Equality Strategy 2018–2023, which is in full alignment with the concerns raised in the 2021 EU Report on the Gender Equality Index. A risk analysis of the potential obstacles to achieving these objectives notes that *"insufficient steps are taken by stakeholders to mainstream gender equality in their work"*. Ways of mitigating this risk are also recommended [7].

Gender inequality is encountered globally in a wider context in different settings in society ranging sadly from the radical circumstances of female exploitation, violence and oppression [8, 9] to the recordings of gender gap in the right to access education, employment, pay, power, and professional development to say the least [10]. From a comparative cultural perspective, it is apparent that gender equality currently in Cyprus stands at the intersection between the *status quo* of Central and Northern European countries where gender equality is highly prevalent such as Iceland and Norway [11, 12] and that of some Middle Eastern, Asian and South American nations and mostly low income countries where gender inequality is prevalent such as Lebanon, India and Brazil [13–16]. Historically, however, the destiny for females was influenced by the patriarchal culture and females were marginalized and suffered social prejudice and/or placed in gender stereotypes and lived under the shadow of patriarchy [17]. Education was a privilege for the upper class until the mid-1950s and the role of females was to be good wives and mothers working in agriculture in rural areas or as dressmakers or handicraft makers. In the mid-1960s, during significant socioeconomic developments, democratisation, modernization and urbanization, primary and secondary school education became compulsory for both genders. In 1960, only 1% of the population had attained tertiary education and these were mostly males. Now, in accordance with the 2021 World Economic Forum's Global Gender Equality Index, Cyprus ranks 49 out of 156 countries with an equality index for Educational Attainment of 99.8% [10]. Furthermore, from 1960, the time Cyprus gained its independence and 2004, its accession to the EU, gradual legislative and concomitant developments in gender equality policy and national action plans were realized [18]. In this framework, the commitment to formal recognition and intention to act for the advancement

of gender equality, the Cyprus Government established various instrumental bodies to coordinate and oversee the relevant activities. In particular, the Gender Equality Unit of the Ministry of Justice and Public Order plays a fundamental role in the harmonization of EU legislation and recommendations in relation to gender equality matters on the island [19]. Through the regular assessments performed at a global and European level, Cyprus shows progress but has substantial room for improvement in all domains/indexes [6, 10].

Currently, compared to one of its nearby geographical neighbours, Greece, with which it also shares the same language, religion and culture, in the Global Gender Gap Index of 2021 rankings [10], Cyprus has performed better in 3 out of the 4 key dimensions/sub-indexes used to evaluate gender equality at an international level (Economic Participation and Opportunity: rank/score: 72/69.4% vs 83/67.2%; [global mean score: 58.33%]; Educational Attainment: rank/score: 49/99.8% vs 67/99.4%; [global mean: 95.0%]; Political Empowerment: rank/score: 86/17.7% vs 115/12.3%; [global mean: 21.8%) and in the overall score (rank/score 83/70.7% vs 98/68.9% out of 156 countries/global mean: 68%). The dimension where Cyprus lags behind Greece, which is Health and Survival, Greece ranks 107 (score: 96.6%) and Cyprus 140 (score: 96%), indicating that essentially both countries are underperforming in this dimension compared to other countries. Nevertheless, there is more uniformity and a smaller gender gap in all the 156 countries evaluated with an average gender gap of 97.5% [10].

In addition, it has been recorded that in most positions of power and leadership even in countries where gender equality policy and mainstreaming progress has been observed there is still a significant gender gap at the detriment of females [10]. Specific examples include disciplines where females are overrepresented such as pharmacy where only 24% of leadership positions in Australian pharmacy organizations are held by females [20], anaesthesiology at a global level [21] and dermatology in the US [22]. Cyprus is no exception to this trend as indicated above with reported statistical data showing male domination in public and private positions of power and authority, namely in leadership positions [10, 23] in spite of a plethora of public and private organizations operating to instigate equality through National Action Plans there is still substantial room for improvement to reach parity [6]. Other than the purpose of equality with respect to opportunities for all individuals in society, another reason for devising mitigation actions to bridge the gender gap in the work-force and specifically in leadership positions, is the benefits gained based on substantial evidence showing that the engagement of women in the work-force and in positions of authority and leadership leads to economic growth, environmental sustainability and innovativeness [24, 25].

Other surveys orientated towards the examination of academic gender disparity illustrate that there is acute concern over the persistence of female underrepresentation in academic faculty positions in medical specialities with a concomitant underperformance in research productivity and promotion to high leadership ranks [22, 26–28]. Gender differences in research productivity in the aforementioned studies were identified and measured via bibliometric analyses using the number of publications, citations and the Hirsch/H/h-index. A similar gender gap has persisted for the STEMM workforce (Science, Technology, Engineering, Mathematics and Medicine) recording a preponderance of males in these disciplines at a global level with a significant trend for male domination in senior employment and concomitant bibliometric record [29, 30]. Interestingly, in the former of these 2 studies, female domination in publications (60–75%) was observed in midwifery, nursing and palliative care with approximate gender parity (~50%) in the biological sciences (molecular biology, biology, genetics, medical genetics). Furthermore, one study identified bias from science faculty in favour of male students indicating that inequality manifests early in the career path of females [31]. Moreover, the latter study, observed that both male and female faculty demonstrated similar gender bias. A recent longitudinal study focusing on reliable publicly available bibliometric

data representing 83 countries has verified the previous observations of gender disparity in total academic productivity in science over a 55-year interval and subsequently illustrates that a shorter female academic career path (characterized by a higher female dropout rate at every stage of the career path) contributes to gender disparity in total productivity and impact. Cyprus is included in this study and it is apparent that its metrics indicate gender disparity. More specifically, for the Cypriot subject data assessed (30 males & 15 females) it demonstrates a statistically significant gender gap with respect to four indicators: (1) annual and (2) total productivity, [gender gap = -5.4% and –12.8% respectively] (3) total impact and (4) career length [gender gap = -43.8% and –21.9% respectively], with a better performance by male academics [31]. Furthermore, with respect to the average gender gap for all 83 countries, for the 4 aforementioned indicators, Cyprus illustrates a smaller gender gap for total productivity, -12.8% vs –27.38% but a larger gender gap is observed for the other 3 indicators. Furthermore, from the selected international sample of 1,523,002 scientists assessed in this study, for all disciplines, females constituted only 27% of the authors with an 18% representation in engineering, 15% in physics and mathematics, 31% in health sciences, 37% in biology, and a maximum representation of 50% in psychology. It is evident that irrespective of the science category, female scientists are underrepresented compared to male scientists. The authors are justified to conclude overall that there is a need to revisit the issue of gender inequality in academia and in particular in relation to supporting the engagement and retainment of women in science through policy adjustments.

Of particular concern are the underlying reasons for gender disparities in all spheres of society. If these factors are identified, they may in theory and hopefully in practice be amenable to intervention to circumvent gender related biases. A recent viewpoint article dedicated to STEM disciplines critically discusses the available literature in this context providing within this discussion a plethora of 'presumed' biases that explain the gender gaps [32]. In brief, these include (1) assumed differences in innate ability which is deemed highly controversial and essentially negated by available data which illustrate only small or no differences in ability, (2) differences in preferences, values or lifestyle choices which are most often centred on the compassionate nature of females to select low income part-time jobs in order to accommodate family obligations, and (3) explicit and implicit biases which centre on stereotypes that associate males with STEM and leadership roles rather than females. Another study highlights that the 'confidence gap' essentially mirroring the above viewpoint on explicit and implicit bias also affects the pay gap among STEM graduates [33]. It is argued that the cultural beliefs about the 'appropriateness' of men and women in STEM affect self-belief and self-confidence. In relation to females, this culminates in lower self-confidence and lower expectations for salaries and essentially allowing their salaries to be compromised as a result of their so-called lower 'self-efficacy.'

Absence of gender parity unsurprisingly, is also observed in the context of research grant awards, nominations for awards for achievement, prestigious awards such as Nobel Laureates and others [21, 32, 34]. Regarding Nobel prizes, it is postulated through the Bayesian model applied to analyse the observations that the most likely causes are related to family obligations and subsequent leaky pipelines, lack of role models, lower resources leading to lower publications and consequently lower probability for nominations [34]. A striking example of gender bias with respect to grant application funding success, originates paradoxically from the Netherlands, a country which appears in the top ranks of the global gender index (31st) and ranked 57th with respect to economic participation and opportunity [10, 35]. Furthermore, with only 3% of Nobel science awards and approximately 8% of the total Nobel awards having been awarded to females to date, the Nobel committees have acknowledged the gender inequality issue only recently and are in the process of improving their nomination procedures in order to promote gender [36].

The present report focuses on the Cyprus Institute of Neurology & Genetics (CING) one of the island's top medical, research and postgraduate academic institutions, as a model for gender distribution trends with respect to 2 of the aforementioned domains of the 2021 EU Report on the Gender Equality Index, those of work and knowledge as well as the investigation of internal collaborative networks so as to identify and characterize any gender trends from this perspective. Internal collaboration can be described as a concerted investment which channels diverse yet complementary scientific knowledge, technical skills and expertise to address a common research enquiry/goal/hypothesis for the advancement of biomedical discovery, medical service/therapy development. This form of collaboration is concomitantly a catalyst for the fortification of the institute's publication impact, access to research funding, expansion, diversification and ultimate sustainability. In particular, the analysis of big data generated and analysed by multidisciplinary expertise can lead to an exponential rise in novel scientific discovery, publications and even patents [37].

The CING was established in 1990 as a non-profit, bi-communal medical, research and academic centre to offer its services to the community in the fields of neurology, genetics and biomedical sciences. At present (2020), it is armed with a workforce of 210 employees, with several national and international collaborations and a research history captured in hundreds of publications in peer-reviewed journals. Its state-of-the-art facilities, infrastructure, expertise, accredited services [38–40], and highly qualified workforce along with the proper external evaluation and government support, have undoubtedly proven instrumental in the development of its highly specialized service programmes and research productivity. Further to that, it has additional accreditations which attest to its equal opportunity policies such as the Human Resource Strategy for Researches (2012), the Sound Industrial Relations; (2015) and Equality Employer (2015) accreditations [41–43].

Other than the provision of high calibre medical and forensic services, the CING is one of the island's top research and postgraduate academic institutions. It is in fact, one of a kind as no other such centre in Cyprus offers the triad of (a) specialized medical, clinical and forensic genetic services, (b) biomedical research and (c) postgraduate education. The institute is composed of: 9 diagnostic & research laboratories, a bioinformatics department, a mouse facility, a biostatistics unit, 5 neurology clinics, a clinical genetics clinic, a specialized ward for neurology, a physiotherapy unit, a pharmacy, social services office and several supporting services (IT, Engineering, Finance & Administration including Health & Safety and Quality, Human Resources, Research Programs, Education and Supplies Offices) which are mandatory for the daily operations of the CING. A significant number of postgraduate students are also involved in the CING's research endeavour through the postgraduate School of CING, established in 2012.

The aims of this study were twofold. Firstly, in the absence of gender gap studies in organisations in Cyprus offering the triad of research, education and services, we set out to assess gender parity selecting CING as a paradigm due to its threefold approach, it's full accreditation as an organisation and the longitudinal data availability due to it being one of the oldest academic organisations in Cyprus. Secondly, we devised a comprehensive analysis method (combining both descriptive statistics and collaborative network topological metrics) that can be used to assess gender parity and is easily flexible, extendable and transferable in other academic organisations.

The analysis herein, evaluates gender distribution in the different departments, clinics, support services, recruitment intervals/years of service, qualifications, student gender distribution and subsequently discusses these observations with respect to the domains of knowledge and work of the 2020 EU Report on the Gender Equality Index and other relevant studies in this context. In addition, a step beyond descriptive statistics analysis was taken by applying

collaborative network analysis to evaluate the representation of the two genders in a term of collaboration (defined as co-authorship) of researchers at CING. Co-authorship networks are graphs in which the nodes represent authors [44]. In such networks two authors are connected by a link if they have co-authored one or more publications. The analysis of co-authorship networks with graph theory methods can answer a broad variety of questions on how patterns of collaboration vary between subjects/labs and between male/female scientists [44, 45]. Using publicly available data we devised a co-authorship network of currently employed researchers at CING and employed graph theory methods to explore the influence and collaborativeness across gender in the CING co-authorship network.

## Materials and methods

### Data collection and descriptive analysis

The CING Human Resource (HR) Office of the Finance and Administration Department provided anonymous frequency data on employees' gender distribution with respect to the different departments and years of employment (1989–2020). PhD students were excluded from this analysis. The postgraduate school provided anonymous summary data on gender distribution for student entries from 2012–2020. From these data, the gender distributions were compared by chi-square test of independence to assess whether there are more female (or male) employees than one would expect by chance in a certain group (e.g., division, department etc.) based on total number of male and female employees and total number of people in each group. For small sample sizes (below 5 data points) for which chi-square is not accurate, the Fisher's exact test was applied for testing statistical significance in the selected categories (as listed in Table 1). All independence tests were performed using the R language (https://www.R-project.org/) [46]. For this analysis, we stratified CING employees in divisions and departments according to the provided information by the HR Office. Note that laboratory secretarial employees were included in the support services division since they do not engage in research activities. Finally, the CING Research Office provided summary data on the number of awarded research grants from 2010–2020 stratified by gender of the principal investigator. A summary of the methods used in the statistical analysis of the network is listed in Table 2.

Ethical review and participant consent were not required since frequency data were anonymously provided by the human resource office, the research officer and post graduate school. The Scopus IDs of CING researchers were retrieved from the publicly available Scopus database using the CING affiliation, and each Scopus ID was given a new ID number (pseudo-anonymized) for the subsequent analyses.

### CING co-authorship network

**Network construction.**   We constructed a co-authorship network of all the CING employees that have published scientific articles with CING affiliation and are currently employed in the CING. As a source of information for the publications of CING employees we used the Scopus database (www.scopus.com, retrieval date: 17/05/2021). All publications with the Cyprus Institute of Neurology and Genetics affiliation were downloaded from Scopus using the search:

*AFFIL (cyprus AND institute AND neurology AND genetics) AND (LIMIT-TO (AF-ID, "Cyprus Institute of Neurology and Genetics" 60071321))*

We restricted the set of scientific articles using the option filter by the year published up to 2020. The filtered total number of publications was 941.

The list of the current laboratory employees of the CING was also retrieved from the CING HR office and the analysis was performed solely for those employed at the CING by the end of

**Table 1. Methods for gender distribution of employees in the CING section.**

| Section | Analysis | Categories | Statistical Methods |
|---|---|---|---|
| Gender Distribution in the CING Divisions and Departments | Comparison of Gender Distribution in the CING across Divisions | ◦ Research & Diagnostic<br>◦ Clinical Services<br>◦ Support Services | Chi-square test of independence & Percentages |
| | Comparison of Gender Distribution in the CING Research & Diagnostic Division | ◦ Clinical Genetics Clinic<br>◦ Neurology Clinics<br>◦ Biochemical Genetics<br>◦ Bioinformatics<br>◦ Cardiovascular & Forensic Genetics<br>◦ Cytogenetics & Genomics<br>◦ Electron Microscopy & Molecular Pathology<br>◦ Molecular Genetics Function & Therapy<br>◦ Molecular Genetics Thalassaemia<br>◦ Molecular Virology<br>◦ Neurogenetics<br>◦ Mouse Facility | Fisher's exact test of independence & Percentages |
| | Comparison of Gender Distribution in the CING Clinical Services Division | ◦ Nursing<br>◦ Pharmacy<br>◦ Physiotherapy | Fisher's exact test of independence & Percentages |
| | Comparison of Gender Distribution in the CING Support Services Division | ◦ Finance & Administration<br>◦ Engineering<br>◦ IT | Fisher's exact test of independence & Percentages |
| Gender Representation in Research & Diagnostic Services Professional Status | Comparison of Position Rankings between Males & Females | ◦ Head<br>◦ Scientist<br>◦ Neurologist<br>◦ Postdoc<br>◦ Laboratory Scientific Officer (LSO)<br>◦ Senior Laboratory Scientific Officer (SLSO)<br>◦ Lab Assistant | Fisher's exact test of independence & Percentages |
| | Comparison of top two ranks in clinical and research strands | ◦ Head Neurologist<br>◦ Neurologist<br>◦ Head Scientist<br>◦ Scientist | Percentages |
| | Comparison of PhD & MD Qualifications between Males & Females | ◦ PhD Holders<br>◦ MD Holders<br>◦ Both PhD & MD Holders | Percentages |
| Gender Relation to Years of Service | Comparison of Years of Service between Males & Females/Recruitment Male: Female Gender Ratio at 4 Year Intervals | ◦ < 4 years of service<br>◦ 5–9 years of service<br>◦ 10–14 years of service<br>◦ 15–19 years of service<br>◦ 20–24 years of service<br>◦ 25–28 years of service<br>◦ 29–32 years of service | Chi-square test of independence & Percentages |
| Gender Distribution of Postgraduate Students in the CING | Postgraduate Student Gender Distribution | Annual entries since the school started operation 2012–2020 | Chi-square test of independence & Percentages |

2020 and focused on the employees within the Research and Diagnostic division. The final list was curated for authors who had duplicate Scopus IDs for whom we merged their publication record for multiple IDs in a single ID. Note that for 15 of the Research and Diagnostic division employees no Scopus ID was found (due to either lack of scientific publications or lack of a Scopus ID due to recent publications that were not yet added to the Scopus database), hence they were excluded from the network analysis. Four more authors were excluded as they had no publications with the CING affiliation up to 2020. Also, PhD students were excluded from the co-authorship network. The final list of authors was pseudo-anonymized for further analysis.

**Table 2. Methods for collaborativeness in CING co-authorship network section.**

| Section | Analysis | Categories | Statistical Methods |
|---|---|---|---|
| **Collaborativeness in CING Co-authorship Network** | Comparison Analysis of network gender stratified network centralities (Degree, betweenness, closeness, eigenvector) | Male vs Female | Two-sample Mann-Whitney-Wilcoxon test |
| | | | One-tailed Mann-Whitney-Wilcoxon test |
| | Assortativity | Male vs Female | Assortativity test |
| | (Degree, betweenness, closeness, eigenvector) | Variables *gender (Male vs Female)* and *head (head vs non-head)* | Two-way ANOVA analysis |
| | Degree | ◦ Male Head | Post-hoc test (Pairwise t-test and ANOVA) |
| | Betweenness | ◦ Male non-Head | |
| | Closeness | ◦ Female Head | |
| | | ◦ Female non-Head | |
| | Betweenness | ◦ Male Head | Post-hoc test (Pairwise Wilcoxon test and Kruskal-Wallis test) |
| | | ◦ Male non-Head | |
| | | ◦ Female Head | |
| | | ◦ Female non-Head | |

**Network analysis.** We used the igraph package in R (http://igraph.org/r/) [47] to process the data and to create, analyse and visualise the co-authorship network. For the visualization of the co-authorship network we used the Fruchterman Reingold graph drawing algorithm included in the igraph package in R. The co-authorship network was further analysed with respect to gender. The author's collaborativeness and influence within the CING co-authorship network was assessed by calculating the following centralities: (1) **degree** which quantifies the number of nodes each node of interest is connected with, (2) **betweenness** which quantifies the number of shortest paths between two nodes going through each node, (3) **closeness** which is a measure of the sum of the length of the shortest paths between the node and all other nodes in the network, and, (4) **eigenvector**, which measures the influence of a node within a network (a high eigenvector score indicates that the node is connected to many nodes with high scores themselves). Also, we computed the **assortativity coefficient** which is positive if similar nodes (based on some property such as gender or department affiliation) tend to connect to each other, and negative if the converse is true.

**Network statistical analysis.** The statistical tests for the network centralities stratified based on gender were performed in R. Normality was tested using the Shapiro-Wilk test method. For the centralities that at least one group was found to be significantly different from the normal distribution, the Matt-Whitney-Wilcoxon test (which does not assume normality of the data as t-test) was used to test whether there was a significant difference in their means. For the centralities that both groups were found not to be significantly different from the normal distribution, the t-test was used to test whether their means were significantly different. We also compared the variances of the two groups for each variable using the F-test and included the result (whether the variances were or were not significantly different) as an input parameter to both the t-test and the Matt-Whitney-Wilcoxon test. A summary of the methods used in the statistical analysis of the network is listed in Table 2.

The distributions of the two GENDER-stratified distributions of each centrality were illustrated using the raincloud plot with boxplots [48]. The raincloud plot was used to optimally visualize raw data, including the probability density and the relevant boxplot.

For the ANOVA (ANalysis Of VARiance) analysis the centrality data for degree, closeness and eigenvector were transformed with log10 transformation to satisfy the normality condition (log10(x) for degree, closeness and eigenvector and log10(x+1) for betweenness). Two-

way ANOVA was performed with gender and seniority (head vs non-head) with the aov function in R (interaction of the two independent variables was also performed). For the ANOVA analysis, the f-value, degrees of freedom, and the p-values were reported for each independent variable. The betweenness centrality data did not satisfy the normality assumption despite the logarithmic transformation. Thus, we used Kruskal-Wallis (a non-parametric alternative to one-way ANOVA test) to test the effect of rank and gender and the pairwise Wilcoxon test to compare group levels pairwise with corrections for multiple testing (using the Benjamini & Hochberg (BH) p-value adjustment method). For the analysis and visualisation of the results, we used the following additional R packages: dplyr [49], tidyr [50], ggplot2 [51], ggpubr [52], RColorBrewer [53], gridExtra [54], cowplot [55], readr [56], rstatix [57].

# Results

## Gender distribution of employees in the CING

**Gender distribution in the CING divisions and departments.** We initially compared the representation of the two genders in the CING across the three main divisions, i.e. (a) Clinical Services, including Neurology and Clinical Genetics Clinics, Nursing, Physiotherapy and Pharmacy employees, (b) Research & Diagnostic, including Molecular Genetics and Clinical Sciences Laboratories (the research subsection of the clinics), and (c) Support Services, including Finance & Administration Engineering and IT departments. As illustrated in Fig 1A, in all three divisions there is a greater proportion of female employees compared to male employees; 77.78% vs 22.22% in Clinical Services and 63.57% vs 36.43% in Research & Diagnostic and 59.26% vs 40.74% in Support Services, (see S1 Table for group numbers). Also, the female to male gender representation was higher in the total CING employees count across all divisions (64.29% vs 35.71%). A chi-square test of independence showed that there was no significant association ($x^2(2) = 2.7641$, p = 0.2511) between gender and the division that an employee works in. The analysis supports the conclusion that gender and division affiliation are not associated with each other and the differences reported in male to female frequencies across divisions were not significant.

Following the analysis across divisions, we assessed the gender balance in the departments for each division separately. The summary data across the CING laboratories is illustrated in Fig 1B (S2 Table). The preponderance of female scientists was apparent in the majority of the CING Molecular Genetics and Clinical Sciences laboratories (9/13). A Fisher's exact test of independence showed that there was no significant association (p-value = 0.4689) between gender and the laboratory that an employee works in. The analysis supports the conclusion that gender and laboratory affiliation are not associated with each other and the differences reported in male to female frequencies across laboratories are not significant.

The analysis of gender distribution within the Clinical Services of the CING indicates that there is over-representation of females across all three departments (Nursing, Pharmacy and Physiotherapy) as illustrated in Fig 1C (see S3 Table for group values). No male employees were employed in the pharmacy department and there was a greater proportion of female employees in the nursing department (males 40% vs females 60%) and the physiotherapy department (males 20% vs females 80%). A Fisher's exact test of independence showed that there was no significant association (p-value = 0.7381) between gender and the department that an employee works in within the Clinical Services division. The analysis supports the conclusion that gender and department affiliation within the Clinical Services are not associated with each other and the differences reported in male to female frequencies across them are not significant.

The comparative analysis of gender distribution within the Support Services departments of the CING (IT, Engineering and Finance & Administration) indicated that there is under-

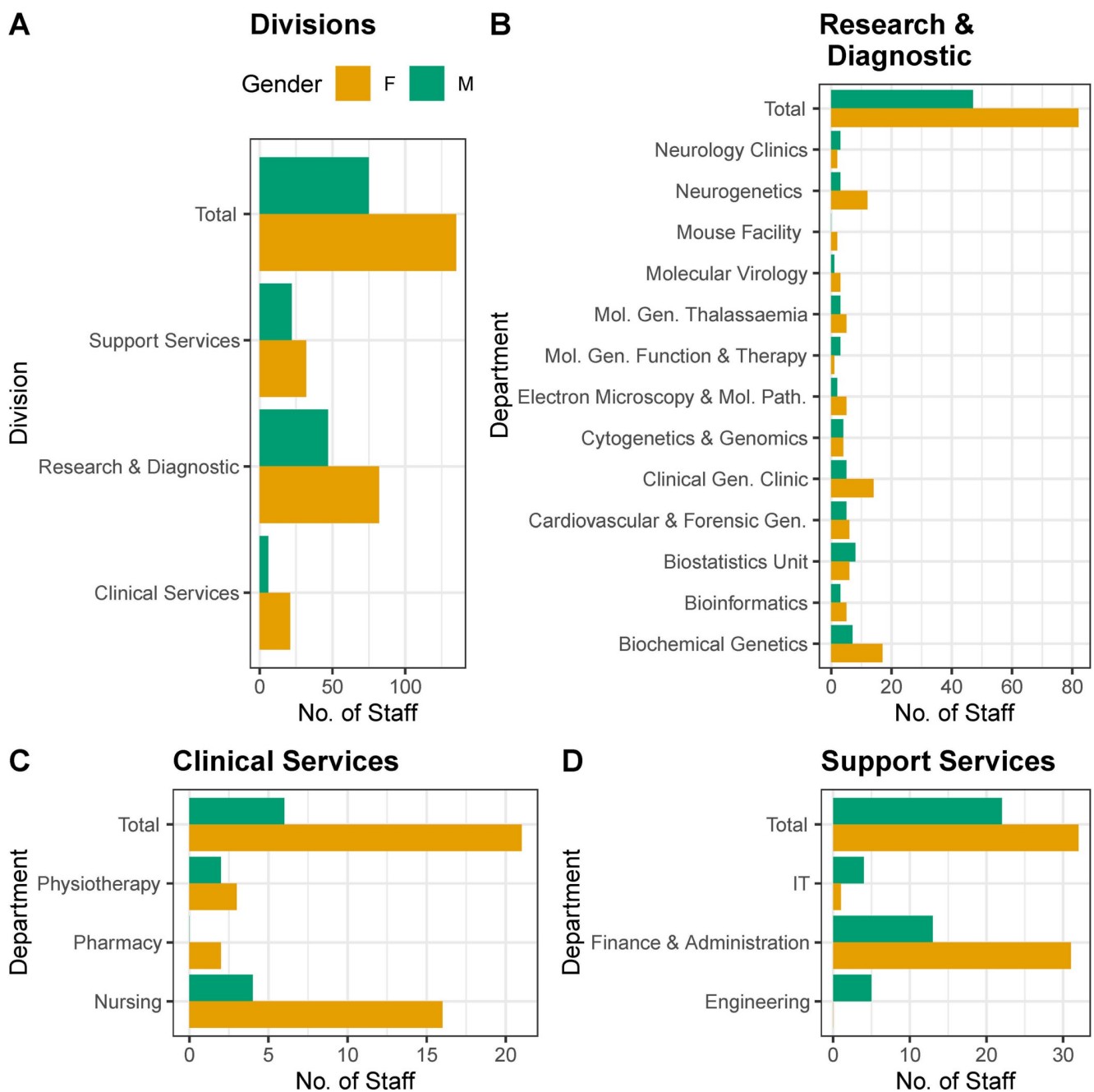

**Fig 1. Gender distribution of employees in the CING. A.** Histogram plots of the gender distribution of employees in the three divisions (Clinical Services, Research & Diagnostic and Support Services) in the CING. **B.** Histogram plots of the gender distribution of employees in the CING Departments (Clinics, Laboratories and Facilities). **C.** Histogram plots of the gender distribution of employees in the Clinical Services departments. **D.** Histogram plots of the gender distribution of employees in the Support Services departments of the CING.

representation of females to males in the Engineering (100% males) and IT departments (20% vs 80%) in contrast, in the Finance & Administration Department, there are more females compared to males (70.45% vs 29.55%) as illustrated in Fig 1D (see S4 Table for group values). A Fisher's exact test of independence showed that there was significant association between gender and department membership in the Support Services division (p-value = 0.0006922).

The analysis supports the conclusion that gender and Support Services department affiliation are associated with each other and the differences reported in male to female frequencies across them are significant.

**Gender representation in research & diagnostic services professional status.** Following the division and subdivision analysis we focused on the largest division, i.e. the Research & Diagnostic Services division to further investigate gender representation in professional status in the CING with respect to (a) their position ranking within the Institute hierarchy and (b) those with top qualifications (either PhD or MD or both). As illustrated in Fig 2A (see S5 Table for group values) gender parity is evident from an overall comparison of senior positions such as Lab & Clinic Directors (Heads) which are held by Senior Scientists and/or Clinicians (50% vs 50%), and Senior Laboratory Scientific Officer (SLSO) (50% vs 50%). In contrast, gender disparity is evident in the other 2 middle ranks with fewer females compared to males: Scientists/Clinicians (26.67% vs 73.33%), Senior Associates (0 vs 100%). The opposite trend was observed for low ranking positions where females were overrepresented compared to males, i.e. for Associate Scientist (65% vs 35%), post-doctoral researchers (postdoc) (75% vs 25%), Laboratory Scientific Officer (LSO) including clinical genetic counsellors (75.93% vs 24.07%) and lab assistant (83.33% vs 16.67%), Hence, although females were overrepresented compared to males in the cumulative number of all the aforementioned groups (63.57% females vs 36.43% males) looking into the different categories reveals that there are more females in low ranking positions whereas there are more males in some of the middle positions. A Fisher's exact test of independence showed that there was significant association between gender and position ranking (p-value = 0.006385). A further investigation of the composition of the two top ranks (see Fig 2B and 2C) stratifying in terms of the clinical (Head Neurologist/Neurologist) and research (Head Scientist/Scientist) strands was carried out. For the clinicians, there was a slight surplus of males to females (3 M & 2 F) for the top rank which was similar for the lower rank (2 M & 1 F). On the contrary, for the research strand there was a slight surplus of females compared to males (4M & 5F) which was reversed for the lower rank (9M & 3F).

A further comparison of qualifications stratified for gender also indicated that out of all PhD holders, females exceeded males (60.56% vs 39.44%) and out of all MD holders, males exceeded females (33.33% vs 66.67%) (for group values see S6 Table).

**Gender relation to years of service.** The recruitment male: female gender ratio over a 32-year period and the years of service was quantified for the employees currently employed in the CING per 4-year interval, as illustrated in Fig 3 (S7 Table). We found a majority of female recruits at every time interval which is concordant with female domination in pooled CING employees. A chi-square test of independence showed that there was no significant association between gender and years of service ($x^2(7) = 4.3838$, p = 0.7347). The analysis supports the conclusion that gender and entry year interval are not associated with each other and the differences in male to female frequencies across them are not significant.

## Gender distribution of postgraduate students in the CING

To test our hypothesis for a greater interest exhibited by females to pursue the biological sciences as one of the reasons for subsequent over-representation in the field, we assessed the annual and overall ratio of female and male students in CING for their postgraduate studies. Firstly, as shown in Fig 4 (S8 Table), at all academic year entries, there were more females than males. Overall the representation of female students in the school throughout its operation was 78% vs 22% of male students. A chi-square test of independence showed that there was no significant association between gender and year of entry for students ($x^2(7) = 13.567$, p = 0.05945). The analysis supports the conclusion that gender and entry year are not

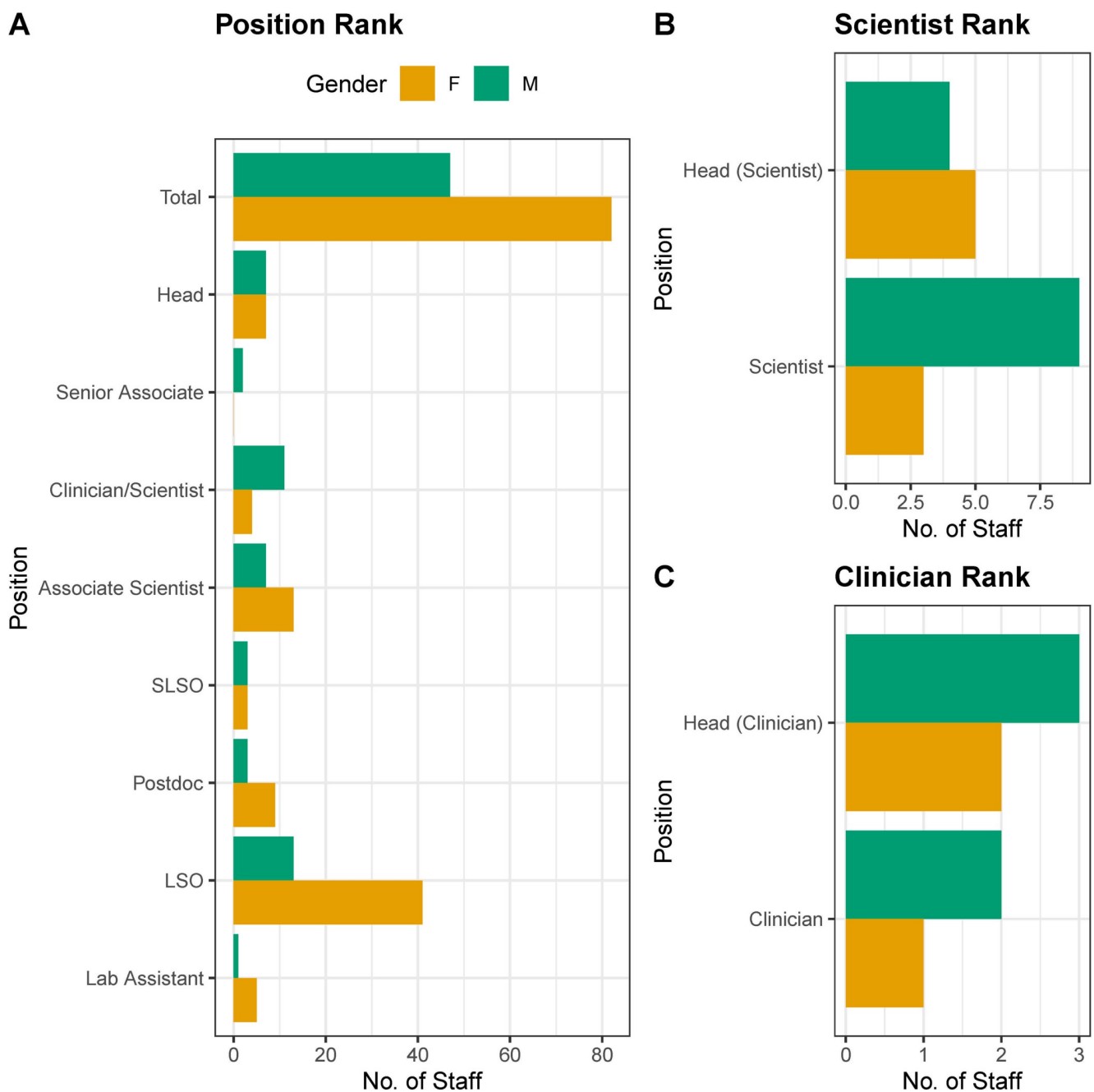

**Fig 2. Gender representation across positions in the CING. A.** Histogram plots of the gender distribution of employees in position rankings in the Research & Diagnostic division in the CING. **B**. Histogram plots of the two top ranks in the scientific strand. **C.** Histogram plots of the two top ranks in the clinical strand.

associated with each other and the differences in male to female frequencies across them are not significant.

## Collaborativeness in CING co-authorship network

Following the descriptive analysis of the gender-based employees' and postgraduate students' representation in the CING, we focused on the exploration of the intra-institute

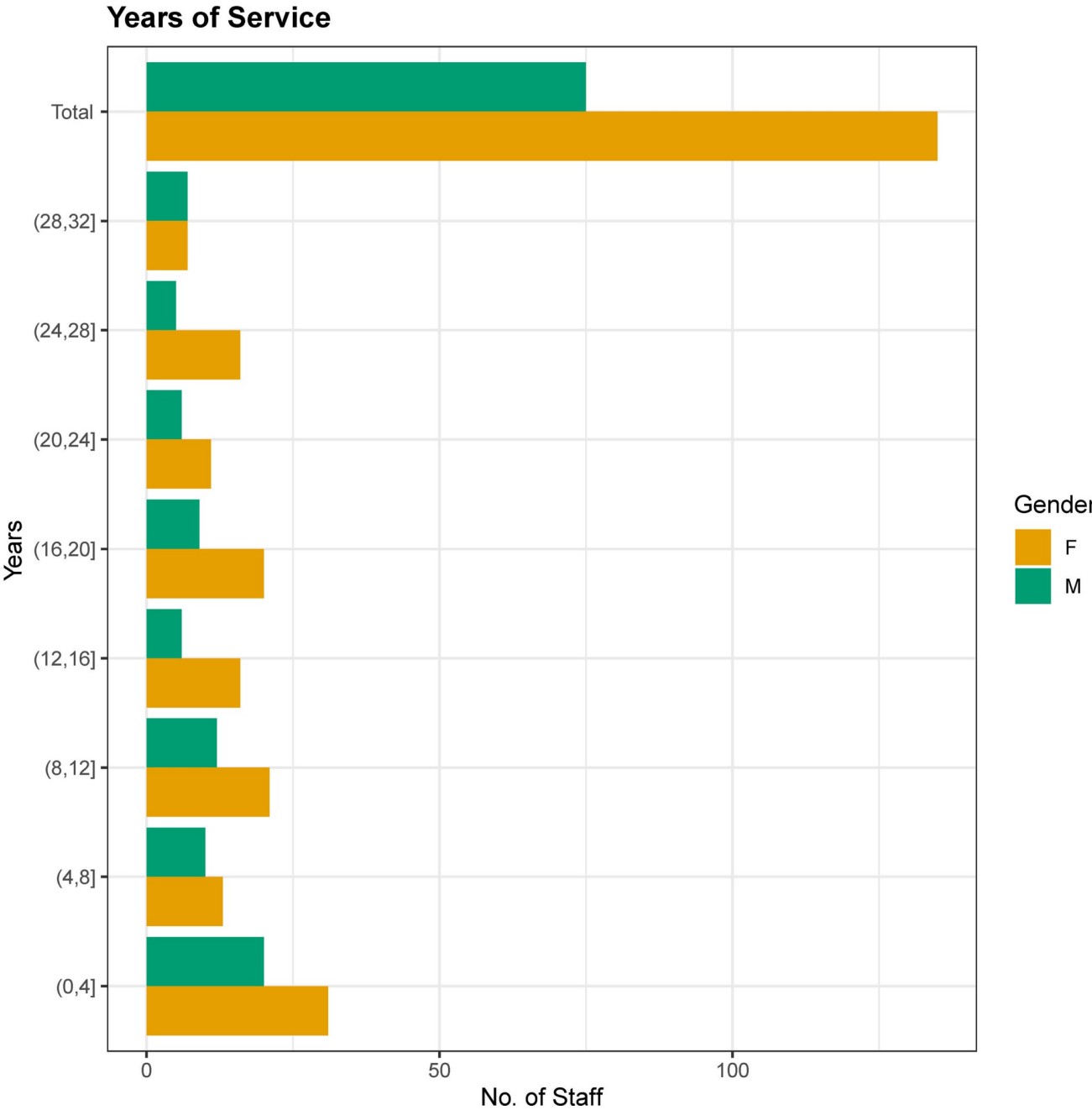

**Fig 3. Comparison of years of service between males & females.** Histograms of recruited employees per 4-year interval.

dynamics among researchers in terms of gender. We generated a co-authorship network (110 nodes, 71 females and 39 males) for the currently employed researchers in CING with publications up to 2020, in order to explore collaborativeness and influence of each gender. The generated co-authorship network was fully connected except two nodes/authors who only co-authored papers among themselves, as illustrated in Fig 5 (network edjelist is available at S9 Table). Note that the two isolated authors were employed recently in CING (2019–2020).

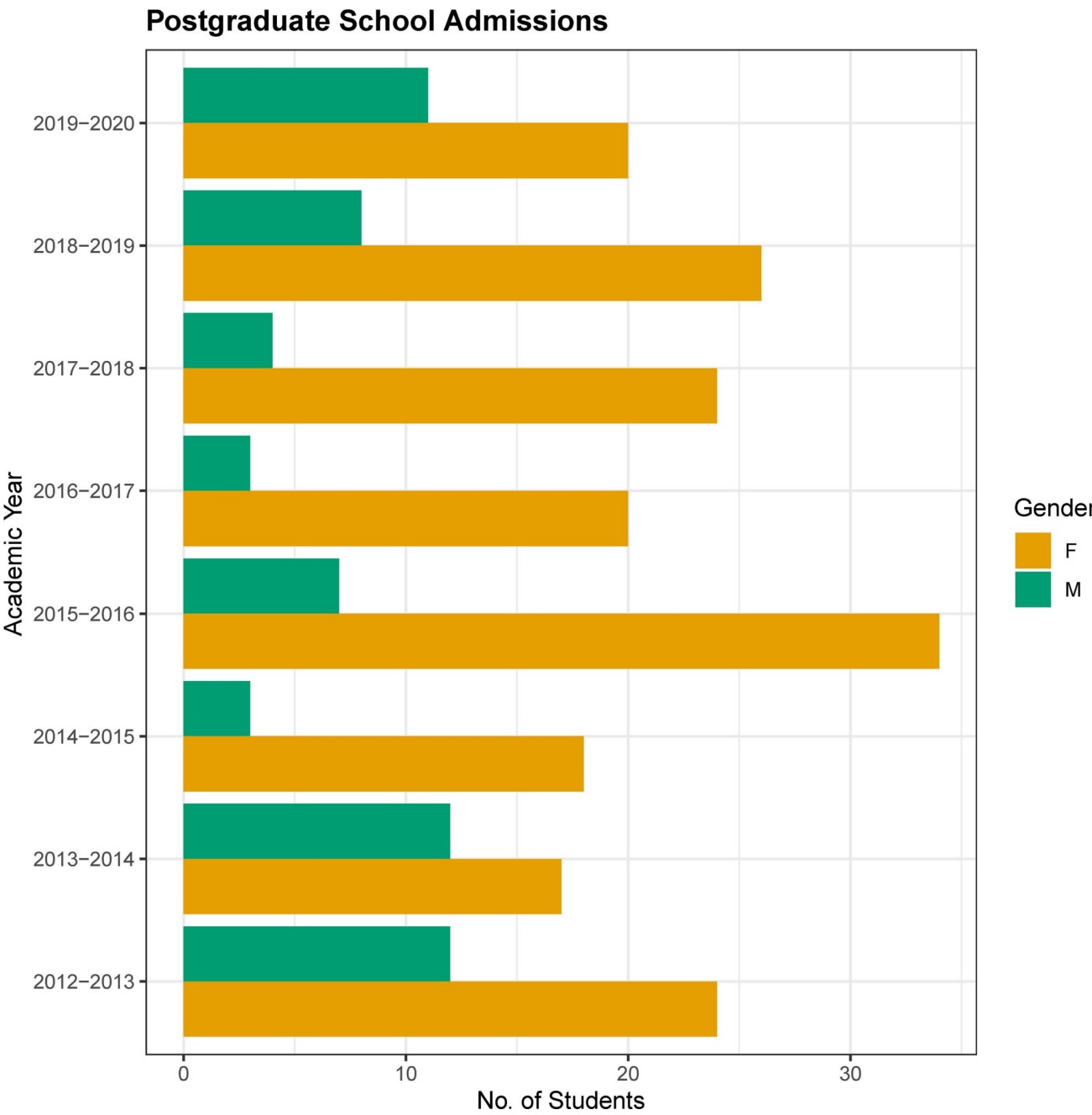

**Fig 4. Successive academic year postgraduate student gender representation.** Histogram of the percentage of female and male students recruited in across 8 consecutive academic years.

To further assess network metrics, we selected the largest connected subcomponent of the co-authorship network. We computed the degree centrality which quantifies in this case for each author, the number of other authors with whom he/she has co-authored at least one paper. By carrying out a two-sample Mann-Whitney-Wilcoxon test we found that males' average degree was significantly different than females' average degree with a p-value = 0.005491 (see Fig 7A). By further testing with one-tailed Mann-Whitney-Wilcoxon test we concluded that the males' average degree was significantly greater than females' average degree (p-

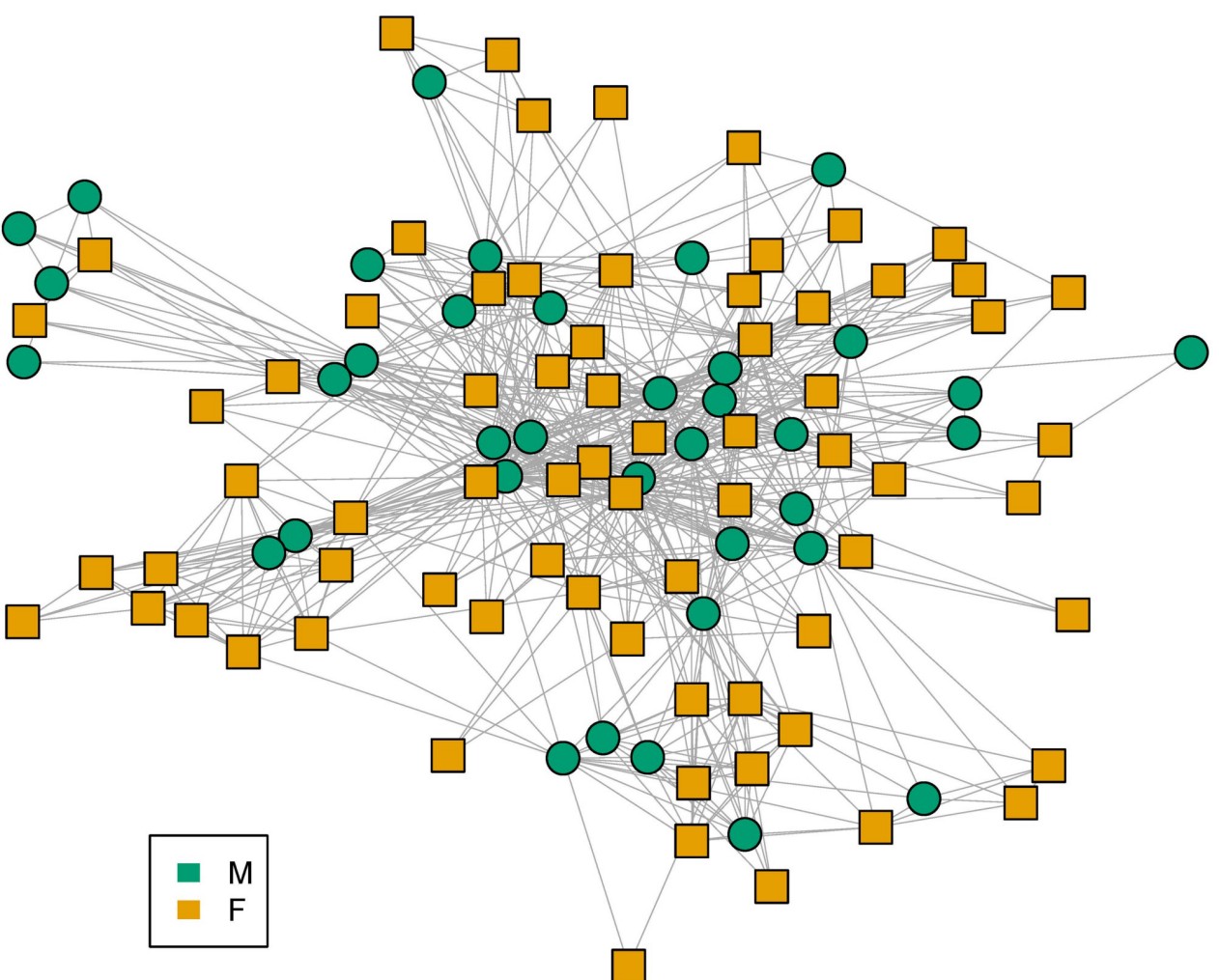

**Fig 5. Gendered co-authorship network of CING employees.** Circle nodes indicate male authors, whereas square nodes indicate female authors. A link between two nodes/authors indicates co-authorship of at least one research paper.

value = 0.0027). We also explored whether authors are more preferentially connected to authors of the same gender. For this we computed the assortativity coefficient (positive if similar vertices, based on some property tend to connect to each other and negative if they do not) of the network with respect to the two groups. We found that the CING co-authorship

network has a negative assortativity score with respect to the two different gender groups (-0.02885) indicating that neither male nor female authors preferentially collaborate based on gender. In contrast, when assortativity is calculated with respect to the department the score is positive (0.2834) indicating that researchers are more likely to publish with members of their lab, as intuitively expected. The co-author network with authors clustered based on their laboratory/clinic affiliation is illustrated in Fig 6 (network edjelist available in S10 Table). We further checked whether there was a preferential co-authorship trend with respect to gender by computing separately for the two groups the degree for each node to only females and to only males (see Fig 7B and 7C). We concluded that for the group of males, their average degree of connection to only males were not significantly different from their degree of connection to only females (p-value = 0.318). Equivalently, we concluded that for the group of females their average degree of connection to only males were not significantly different from their degree of connection to only females (p-value = 0.1933).

Secondly, we computed the betweenness centrality which quantifies the number of times an author lies on the shortest path between other authors. By carrying out a two sample Mann-Whitney-Wilcoxon test we found that males' average betweenness was significantly different than females' average betweenness with a p-value = 0.0060395 (see Fig 7A). By further testing with one-tailed Mann-Whitney-Wilcoxon test we concluded that males' average betweenness

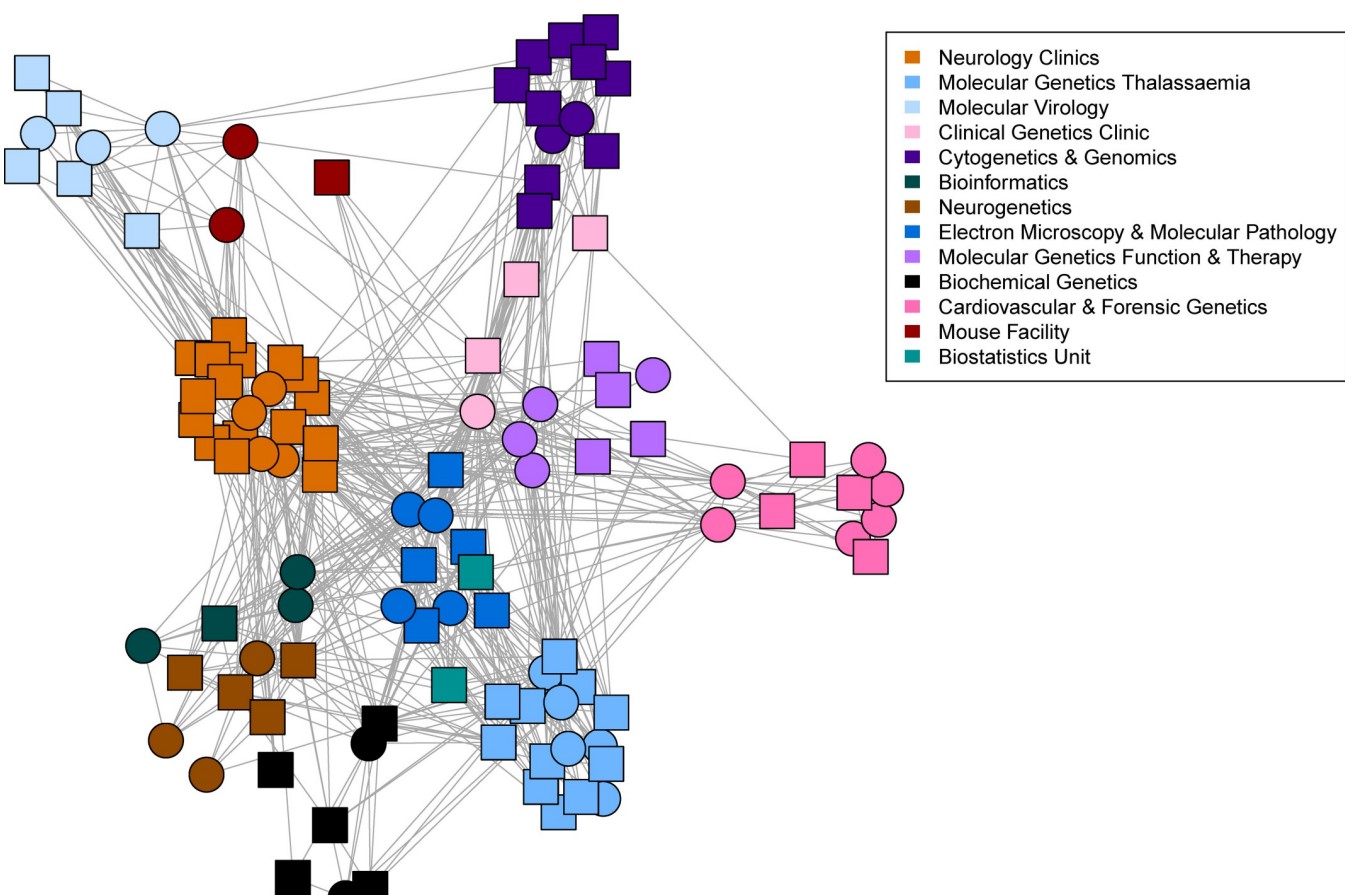

**Fig 6. Departmental co-authorship network of CING employees.** Square nodes indicate female authors and circle nodes indicate male authors. Node colour indicates department affiliation whereas a link between two nodes/authors indicates co-authorship of at least one research paper.

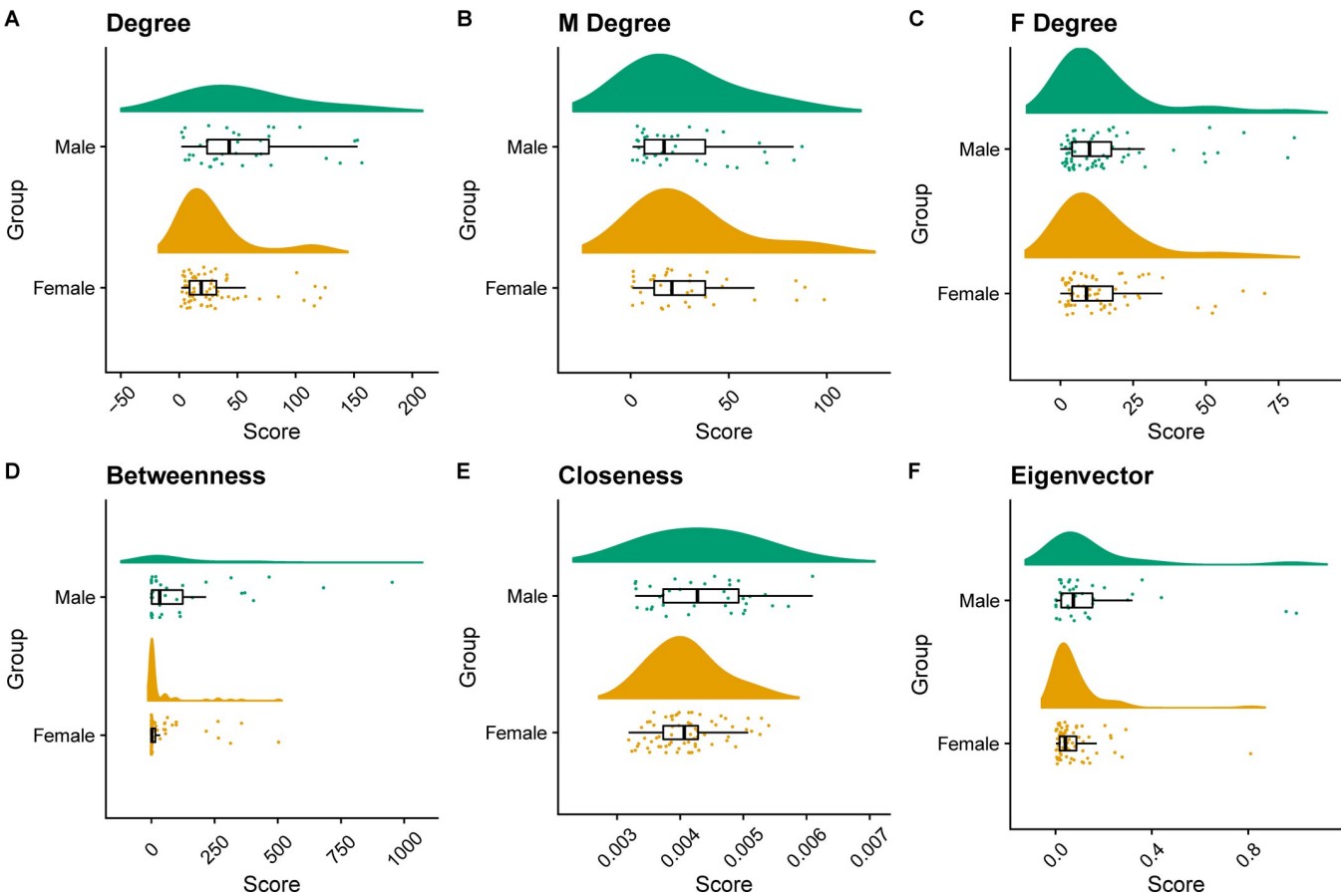

**Fig 7. Raincloud plots for the male/female group centralities.** The raincloud plots with boxplots for the scores of the computed centralities for the female and male author groups in the CING co-authorship network including the average (1) degree for all (2) degree for male authors (M Degree) to male/female authors, (3) degree for female authors (F Degree) to male/female authors, (4) betweenness, (5) closeness and (6) eigenvector centrality.

was significantly greater than females' average betweenness (p-value = 0.003019). Thirdly, we computed the closeness degree which quantifies how close an author is to all other authors in the network. By carrying out a two sample Mann-Whitney-Wilcoxon test we found that males' average closeness was not significantly different than females' average closeness (p-value = 0.1241). Lastly, we computed the eigenvector centrality which quantifies the reaching influence of a node in a network. By carrying out a two sample Mann-Whitney-Wilcoxon test we found that males' average eigenvector centrality was significantly different than females' average eigenvector centrality (p-value = 0.04546). By further testing with one-tailed Mann-Whitney-Wilcoxon test we concluded that males' average eigenvector centrality was significantly greater than females' average eigenvector centrality (p-value = 0.02273).

A key observation was that the top influential nodes in terms of centralities were mostly highly ranked employees (heads/senior associates), as listed in Table 3 with the top 25% quantile for each calculated centrality. To further explore the role of seniority in shaping authors' influence in the network, we performed two-way ANOVA analysis by both the variable *gender* and the variable *head* (Heads vs non-Heads) see S9 Table. Note that Senior associates were included in the Heads group as they essentially represent a class of employees which were head of departments and are now emeritus employees. We found a statistically significant difference in average degree by both *gender* (f(1) = 12.83, p < 0.001) and by *head* (head vs non head) (f(1)

**Table 3. CING co-authorship network centralities.** The top quantile scored authors (N = 27) sorted based on the centralities degree (DEG), betweenness (BTW), closeness (CLS) and eigenvector (EIG). The authors' gender (G) is included (F/M) as well as the annotation whether they are part of the Head (H) group with yes (Y) or no (N).

| ID | DEG | G | H | ID | BTW | G | H | ID | CLS | G | H | ID | EIG | G | H |
|---|---|---|---|---|---|---|---|---|---|---|---|---|---|---|---|
| 5 | 51 | M | Y | 5 | 866.309726 | M | Y | 5 | 0.00609756 | M | Y | 5 | 1 | M | Y |
| 101 | 41 | M | Y | 101 | 601.025174 | M | Y | 101 | 0.00571429 | M | Y | 101 | 0.77188497 | M | Y |
| 90 | 36 | M | Y | 90 | 473.000207 | M | Y | 90 | 0.00546448 | M | Y | 86 | 0.75289375 | F | Y |
| 86 | 35 | F | Y | 104 | 399.614659 | M | Y | 86 | 0.00540541 | F | Y | 90 | 0.72562001 | M | Y |
| 31 | 33 | M | N | 95 | 383.899932 | F | Y | 31 | 0.00534759 | M | N | 31 | 0.69134467 | M | N |
| 95 | 30 | F | Y | 86 | 331.008427 | F | Y | 100 | 0.00529101 | F | Y | 20 | 0.63712085 | F | N |
| 96 | 29 | M | Y | 98 | 327.237835 | M | Y | 20 | 0.00526316 | F | N | 100 | 0.61856081 | F | Y |
| 104 | 28 | M | Y | 96 | 326.279208 | M | Y | 95 | 0.00512821 | F | Y | 34 | 0.61236901 | M | Y |
| 100 | 28 | F | Y | 31 | 288.109606 | M | N | 93 | 0.00510204 | M | N | 30 | 0.59252726 | M | Y |
| 20 | 27 | F | N | 91 | 241.275015 | M | N | 37 | 0.00507614 | F | N | 93 | 0.57738678 | M | N |
| 34 | 27 | M | Y | 69 | 226.283038 | F | Y | 96 | 0.00505051 | M | Y | 96 | 0.56524883 | M | Y |
| 92 | 24 | F | Y | 92 | 185.154143 | F | Y | 92 | 0.00505051 | F | Y | 104 | 0.55348565 | M | Y |
| 30 | 24 | M | Y | 2 | 176.492417 | M | N | 104 | 0.00502513 | M | Y | 37 | 0.53967297 | F | N |
| 98 | 23 | M | Y | 100 | 172.783835 | F | Y | 34 | 0.005 | M | Y | 95 | 0.52088075 | F | Y |
| 18 | 23 | F | N | 77 | 170.794124 | F | Y | 21 | 0.005 | M | N | 16 | 0.51062491 | M | N |
| 37 | 23 | F | N | 21 | 150.532292 | M | N | 98 | 0.00492611 | M | Y | 98 | 0.48767966 | M | Y |
| 93 | 23 | M | N | 18 | 146.393773 | F | N | 18 | 0.00487805 | F | N | 92 | 0.47764345 | F | Y |
| 2 | 22 | M | N | 20 | 143.541169 | F | N | 77 | 0.00483092 | F | Y | 14 | 0.47556093 | M | N |
| 77 | 22 | F | Y | 34 | 138.655663 | M | Y | 14 | 0.00480769 | M | N | 74 | 0.47299052 | F | N |
| 21 | 22 | M | N | 33 | 134.536181 | M | N | 30 | 0.00478469 | M | Y | 21 | 0.46789572 | M | N |
| 16 | 21 | M | N | 37 | 116.021488 | F | N | 91 | 0.00478469 | M | N | 18 | 0.45830329 | F | N |
| 74 | 21 | F | N | 19 | 103.453974 | F | N | 70 | 0.0047619 | M | N | 15 | 0.42536756 | F | N |
| 91 | 20 | M | N | 66 | 102.248261 | F | N | 11 | 0.0047619 | F | Y | 91 | 0.4149967 | M | N |
| 69 | 20 | F | Y | 16 | 91.1682164 | M | N | 19 | 0.00473934 | F | N | 70 | 0.40841616 | M | N |
| 70 | 19 | M | N | 30 | 81.7609686 | M | Y | 16 | 0.00471698 | M | N | 19 | 0.4038118 | F | N |
| 14 | 19 | M | N | 93 | 80.7629666 | M | N | 74 | 0.00471698 | F | N | 11 | 0.40038986 | F | Y |
| 19 | 18 | F | N | 70 | 77.1641899 | M | N | 2 | 0.00469484 | M | N | 77 | 0.39868518 | F | Y |

= 49.08, p<0.001). The interaction among *gender* and *head* was not found to be statistically significant (f(1) = 0, p = 0.9966). In terms of the average closeness we found a statistically significant difference by both *gender* (f(1) = 5.483, p < 0.05) and by *head* (head vs non head) (f(1) = 54.004, p<0.001). The interaction among *gender* and *head* was not found to be statistically significant (f(1) = 0.165, p = 0.6852). In terms of the average eigenvector centrality we found a statistically significant difference by both *gender* (f(1) = 5.495, p < 0.05) and by *head* (f(1) = 39.288, p<0.001). The interaction among *gender* and *head* was not found to be statistically significant (f(1) = 0.215, p = 0.644). For the average betweennesses, we found by computing the Kruskal Wallis test statistically significant differences for both *gender* (chi-squared = 7.5564, df = 1, p-value <0.01) and for *head*: chi-squared = 33.949, df = 1, p-value < 0.0001).

A post-hoc test (pairwise t-test and ANOVA) revealed that for the centralities Degree, Closeness and Eigenvector, the groups whose means were statistically different from one another were (1) female heads (HF) to female or male non-heads (NHF/NHM) and (2) male heads (MH) to female or male non-heads (see Fig 8). The same was observed for the betweenness centrality which was tested with pairwise Wilcoxon test and Kruskal-Wallis test, a nonparametric alternative method (see Fig 8).

Overall, these network analysis results show that although females exceeded males in the co-authorship network, in line with the overall abundance of female to male employees in the

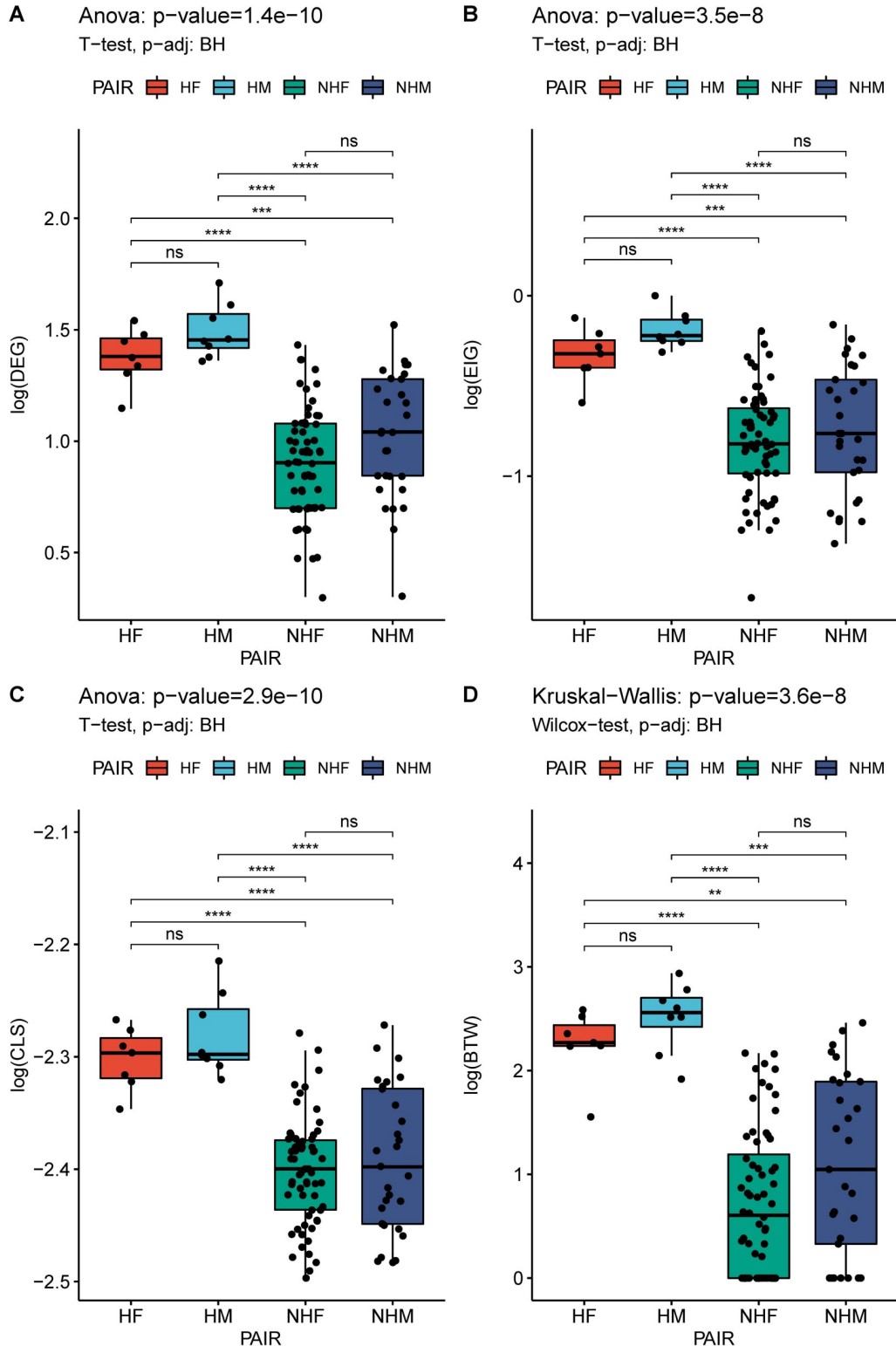

**Fig 8. Post-hoc analysis for network centralities. A-C.** Post-hoc test (pairwise t-test and ANOVA) for the centralities Degree (DEG), Closeness (CLS) and Eigenvector (EIG), the groups whose means were statistically different from one another. The groups were (1) female heads (HF) to female non-heads (NHF), male heads (HM) and male non-heads (NHM). **D.** Post-hoc results for the betweenness (BTW) centrality which was tested with a non-parametric alternative method (pairwise Wilcoxon test and Kruskal-Wallis test). The significance of the p-value in each test in indicated with the following symbols: $< 0.0001$ '****', $< 0.001$ '***', $< 0.01$ '**', $< 0.05$ '*', $< 1$ 'ns'.

CING, the most influential nodes in terms of collaborativeness were predominantly males with regards to average degree, betweenness and eigenvector centralities but not with regards to closeness. However, the factors that modulated the magnitude of these centralities where both the gender and the seniority (head vs non-head) of the authors indicating an additive relationship among the two. The post-hoc analysis shows that significant between-group differences were for both males to females when belonging in different ranks but not when belonging in the same rank and for same gender employees when belonging to different ranks.

## Discussion

The present investigation is to our knowledge the first to address gender parity in a Cypriot institute engaged in specialized service provision, research and postgraduate education. The analyses herein have succinctly demonstrated female over-representation at the CING.

First, through descriptive statistics stratified for gender in different comparative analyses focusing on the various CING divisions, departments, clinical services and support services (Fig 1 and S1–S4 Tables), we observed a consistent greater prevalence in female representation, with the exception of the support services where we observe a statistically significant relation of gender and affiliation to specific division, with a preponderance of females in the Finance & Administration department with the inverse in the IT and Engineering departments. The latter observation agrees with the consensus observation described in the literature and the EU Equality Index [29, 30, 58] on gender distribution amongst the STEMM disciplines. Interestingly, in the nursing arena in the CING, female gender appears to dominate. Here, it would seem that the stereotype role of the female caregiving nurse is in place, in agreement with former observations as well [30].

Second, with respect to positions and ranks in employee hierarchy (Fig 2 and S5 and S6 Tables), we found that the positions ranking, was significantly associated with gender, with more males compared to females in middle ranking positions (i.e. the scientist/neurologist (middle rank) and senior associate (middle rank)) and more females in low ranking positions (i.e. associate scientist, post-doc, LSO and lab assistant ranks), which reflects observations of previous reports [22, 26–28]. These findings may subtly allude to the underlying reasons proposed by these former studies such as the family obligations and career breaks for maternity leave and/or other family obligations or shorter career lengths imposed by various external factors and therefore lead to the application, recruitment and propagation of presence of more females in lower ranking positions in employment hierarchy [29–31]. It is also expected that in the coming years the observed gender distribution with respect to associate scientist and scientist will be adjusted and ultimately reach parity with the promotion of the associate scientists to the scientist position. Notably, a satisfactory observation is equal representation in the top rank of the Institute hierarchy (Heads group).

Third, the evaluation of years of service and concomitant recruitment stratified for gender, highlights a surplus of females overall, and at each 4-year interval since the CING was established in 1990. The past and present CING gender recruitment distribution has remained constant with future fluctuations possible in light of the infiltration of other disciplines (e.g.: mathematics, data science and bioinformatics to explore big data generated from whole genome, transcriptome and proteome analyses) into the field of medical genetics which at present appear to be male dominated. This is a point of interest that should be assessed in the future.

Fourth, the recruitment of postgraduate students from the time the school was founded (2012) until 2020, was again associated with a greater proportion of females compared to males across all academic years. Although, it is a challenge to speculate about the potential

explanations for the observed gender distribution among the different CING departments with limited evidence, a plausible hypothesis based on the present empirical data from the postgraduate student recruits of the CING, is that females have a greater tendency than males to pursue tertiary education in the biomedical sciences and indeed at postgraduate level as already documented [29]. As a result of the latter preference, female applicants per vacancy are greater than male applicants as is the case from our documented experience at recruitment interviews. The latter observation is also in line with recent reports indicating that gender parity exists in the biological sciences publication statistics/bibliometrics [29, 31] but also in accordance with the survey of the EU Equality Index, which indicates that Cyprus has made progress, albeit small (+0.7%) since 2010 in education, referred to as the domain of knowledge [58]. Another potential explanation is that females may be more successful at interview and most importantly that the CING as an egalitarian employer, practices its equal opportunity policy as corroborated by the state accreditations related to equality [41–43]. It is also plausible to speculate, that since there are more low-ranking positions in our employment hierarchy (i.e LSO and Associate Scientist) and females are over-represented overall in the CING, they would be found to occupy these positions at a higher frequency than males. No bias is demonstrated against females in recruitment in contrast to other published surveys [29, 59]. It is also conceivable and probable that the observed distribution of gender at the CING is the product of the simultaneous contribution and interaction of all the above explanations and possibly other social, domestic, cultural and economic influences in concert with personal preferences.

Fifth, the assessment of the degree of collaboration between males and females within the Institute within its co-authorship network pointed to sufficient collaboration intra- and inter-departmentally between males and females as co-authorship was associated with department affiliation and not gender. In terms of the degree of influence, we found that although females exceeded males in the co-authorship network, in line with the overall abundance of female to male employees in the CING, the most influential nodes in terms of collaborativeness were predominantly male. However, with further testing of the influence of gender versus belonging in the highest rank, we found that both factors were statistically important in shaping an author's influence in the co-authorship network. Interestingly, when considering only head vs non-head groups, intergroup differences were not observed for groups of different gender in the same rank. Thus, these findings indicate that as long as career progression is ensured for both genders their equal influence within the collaborative network of the CING is achieved.

In agreement with the 2021 EU Gender Equality Index findings, the CING exemplifies the increment in the domain of knowledge but also work. It is also important to note that in accordance with the Equality Index Report, 38% of females compared to 32% of males in Cyprus attain tertiary education, which is greater than the EU average (26 vs 25% respectively). These statistics agree with the overall female to male gender ratio in the CING. In relation to the domain of work, within the CING environment, females and males have access to equal pay, job security, health & safety, career advancement, training activities, conference attendance, state-of-the-art working infrastructure and resources to fulfil their duties to the best of their potential and the opportunity to continue to develop and participate in lifelong learning. In contrast to the *status quo* with respect to equal pay at the CING, a recent study in New Zealand's Universities reports a very radical gender pay gap as well as in promotion and research funding with preferences for males [47]. This phenomenon is also recorded by the 2019 data of the national science foundation in the US where the median annual salary of male employees with university degrees in all science and technology disciplines is greater for males ($70,000 vs $95,000) with only a minor incremental difference recorded for females in the biological sciences at the start of their careers but towards the end of their careers in the age-group 50–75 years, males earn more ($68,000 vs $101,000) even in this female dominated

discipline (Tables 9–17 in [60]). In addition, female CEOs in not-for-profit US hospitals were shown to earn 22.6% less than their male counterparts [61]. At the CING, it is also important to note, that all scientists with postgraduate degrees irrespective of gender, participate in service provision, research and teaching/supervising/mentoring within the framework of the postgraduate school. In addition, CING PhD/MD holders, irrespective of gender are eligible to apply for research funding. Recent available statistics of grant applications (2012–2020) indicated that 57.89% of awarded grants had male PIs whilst 42.10% had female PIs. In light of all these findings and CING policies it is clearly evident, that the equal opportunity within all the activities of the CING for both genders to participate and contribute prevails.

Unquestionably, the CING has embraced gender equality from the point of its establishment as illustrated in the data presented herein. Our study in juxtaposition to other studies in the medical specialities differs from several perspectives such as the small number of individuals in the upper ranks of the employee hierarchy but also in the very nature of the composition of the workforce which is more diverse (molecular biologists, biostatisticians, bioinformaticians, clinicians, support service employees etc) rather than being composed solely of physicians [22, 26–28].

From a socioeconomic perspective, the observation of significant female representation at the CING points to a concomitant high level of female education, in agreement with the current EU Gender Equality Index, preference for the field, social and economic independence. As a result, the CING benefits from the innovative potential of female scientists working in harmony with male scientists to produce high impact research, specialized services and fulfil academic duties in its postgraduate school. In light of this assessment focusing only on basic demographic data, it is evident that the CING exemplifies a modern egalitarian institute which has succeeded in its goals and has not indulged in a conscious bias of specific gender recruitment in any position. This is an important social development contributing to the promotion of gender diversity in the field of medical research, service provision and delivery of postgraduate education.

Furthermore, in contrast to the CING, recent statistical data published by an equivalent academic and research organization in Cyprus in the context of publicly available policy documents [62] and a technical report on gender equality in STEM at the University of Cyprus (UCY) [63], clearly show male domination in academic leadership, senior academic ranks, senior management and in doctoral students. Also, in agreement with our analysis, female students dominate in the biological sciences department in UCY [63]. However, since these data were not stratified in accordance with the subspecialties within the biological sciences as we have analysed our data, an accurate comparative analysis with the data of that report cannot be performed at this stage. In addition, the priority of the CING is in service provision followed by research and then academic activities in contrast to the local universities whose objectives are in the inverse priority ranking. This may lead to differences due to personal preferences of individuals in job selection. Also, the same preferences for biological sciences have been reported by the US National Science Foundation biennial statistics where more females have been recorded in the biological sciences [64]. Future, similar comparative analyses in other local biomedical and tertiary education institutions may indeed clarify whether the data presented herein, are representative of Cypriot gender equality culture in research, academia and other biomedical professions.

The present study illustrates an informative approach that can be used to study both gender representation and internal collaboration in order to identify weaknesses and strengths in relation to gender equity. Concomitantly, this work revealed an emerging paradigm of how commitment to the implementation of organizational policies for equal opportunities [65] along with EU monitoring and state surveillance through accreditation [41–43] on gender equality

in the working environment have succeeded in securing an equal opportunity working culture in one of the island's oldest biomedical institutions. More specifically, we believe that the combination of policies and regulations formulated and implemented through the CING Human Resource Office and verified by state accreditations for articles stipulated in employment law have contributed to the success of the CING as an egalitarian employer whereby equivalent opportunity for both genders in recruitment, participation and professional development, pay, leave, awards for outstanding performance, grant applications and work-life balance is promoted by a spectrum of policies and regulations. These critical organizational instruments include but are not limited to the following examples: (a) Regulation concerning Staff Recruitment & Selection; (b) Dependent Care Policy (c) Security of Employment (d) Staff Performance Evaluation (e) Civility & Mutual Respect (f) Policy on Prevention and Management of Work-Related Stress (g) Sexual Harassment Policy and Procedure (h) CING Awards Policy and (i) Staff Promotion Scheme. From a cultural perspective, this study exemplifies that females can succeed in a social context that may have been viewed traditionally as a male privilege [17]. It is a paradigm of how policies and social advancements in education and employment can empower women to succeed in a modern society.

## Conclusions

The current study has demonstrated an overall positive outlook in terms of gender parity within the CING. This includes an over-representation of females in the: (1) CING employees across all divisions (2) postgraduate students and (3) recruits across the years of operation of the CING. A plausible explanation as we have shown for this observation is the choice of more female students to pursue postgraduate studies in molecular genetics and ultimately as an employment specialty. Also, with respect to recruitment policy, the CING fulfils the criterion of a modern paradigm of an egalitarian biomedical institute. In addition, the surplus of females in the associate scientist position is expected through the official promotion scheme of our Institute to progress to the scientist position whereby parity at least will be reached in the coming years.

Further demonstration of the CING's success in fostering a gender equality culture in academic collaboration within researchers in the CING is exemplified by the co-authorship network findings. It was shown that researchers were likely to collaborate with each other if they belonged to the same department irrespective of gender. Further comparison of the two groups with respect to their influence in the network in terms of occupying the positions of highest centrality scores, indicated that both gender and seniority level (head vs non-head) were significant in shaping the authors' influence, with no significant difference in those belonging in the same seniority level with respect to their gender. This observation indicates that equal opportunity for career progression culminates in equal academic achievement for both genders. To conclude, our study has validated the formal recognition/accreditation of the CING's policies and procedures pertinent to its egalitarian culture through the majority of metrics of gender equality assessed in this study.

Overall, the implications of this study are twofold. Firstly, we highlight a best practice example in a society with gender disparity across different domains including the local STEM workforce which can be adopted by other organisations in Cyprus and beyond. Secondly, we propose a new approach for evaluating gender parity in terms of influence in collaboration in academic organizations. This type of work is essential in quantifying whether new policies in place result in closing the gap in academic institutions, as required by the EU with the requirement of Gender Equality Plans.

## Supporting information

**S1 Table. Gender distribution in the CING divisions.**
(PDF)

**S2 Table. Gender distribution in the CING departments.**
(PDF)

**S3 Table. Gender distribution in the CING clinical services.**
(PDF)

**S4 Table. Comparison of gender distribution in the CING support service departments.**
(PDF)

**S5 Table. Comparison of ranking positions between males & females.**
(PDF)

**S6 Table. PhD and MD qualifications in the research & diagnostic division.**
(PDF)

**S7 Table. Comparison of years of service between males & females at 4 year intervals.**
(PDF)

**S8 Table. Postgraduate student successive academic year student gender.**
(PDF)

**S9 Table. Network edjelist for the gendered co-authorship network of CING employees.**
(XLSX)

**S10 Table. Network edjelist for the departmental co-authorship network of CING employees.**
(XLSX)

## Acknowledgments

The authors wish to thank and acknowledge the helpful comments on the manuscript by Ms Maria Loizou from the CING Finance and Administration department.

## Author Contributions

**Conceptualization:** Stavroulla Xenophontos.

**Data curation:** Stavroulla Xenophontos, Margarita Zachariou.

**Formal analysis:** Stavroulla Xenophontos, Margarita Zachariou.

**Investigation:** Stavroulla Xenophontos.

**Methodology:** Stavroulla Xenophontos, Margarita Zachariou, Pavlos Polycarpou, Elena Ioannidou, Vera Kazandjian, Maria Lagou, Anna Michaelidou, George M. Spyrou, Marios A. Cariolou.

**Project administration:** Stavroulla Xenophontos, Leonidas Phylactou.

**Software:** Margarita Zachariou.

**Supervision:** Stavroulla Xenophontos, George M. Spyrou, Marios A. Cariolou, Leonidas Phylactou.

**Visualization:** Stavroulla Xenophontos, Margarita Zachariou.

**Writing – original draft:** Stavroulla Xenophontos, Margarita Zachariou, Pavlos Polycarpou, Elena Ioannidou, Vera Kazandjian, Maria Lagou, Anna Michaelidou, George M. Spyrou, Marios A. Cariolou, Leonidas Phylactou.

**Writing – review & editing:** Stavroulla Xenophontos, Margarita Zachariou, Pavlos Polycarpou, Elena Ioannidou, Vera Kazandjian, Maria Lagou, Anna Michaelidou, George M. Spyrou, Marios A. Cariolou, Leonidas Phylactou.

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
