## [Decision Letter · Decision Letter 0]

12 May 2022

PONE-D-21-38904The Cyprus Institute of Neurology and Genetics, an Emerging Paradigm of a Gender Egalitarian OrganisationPLOS ONE

Dear Dr. Xenophontos,

Thank you for submitting your manuscript to PLOS ONE. After careful consideration, we feel that it has merit but does not fully meet PLOS ONE’s publication criteria as it currently stands. Therefore, we invite you to submit a revised version of the manuscript that addresses the points raised during the review process.

We look forward to receiving your revised manuscript.

Kind regards,

Aleksandra Barac

Academic Editor

PLOS ONE

Journal Requirements:

“Margarita Zachariou and George M. Spyrou were funded by the European Commission Research Executive Agency (REA) Grant BIORISE (Num. 669026), under the Spreading Excellence, Widening Participation, Science with and for Society Framework.”

Reviewers' comments:

Reviewer's Responses to Questions

**Comments to the Author**

1. Is the manuscript technically sound, and do the data support the conclusions?

Reviewer #1: Partly

Reviewer #2: Yes

2. Has the statistical analysis been performed appropriately and rigorously? 

Reviewer #1: Yes

Reviewer #2: Yes

3. Have the authors made all data underlying the findings in their manuscript fully available?

Reviewer #1: Yes

Reviewer #2: Yes

4. Is the manuscript presented in an intelligible fashion and written in standard English?

Reviewer #1: No

Reviewer #2: Yes

5. Review Comments to the Author

Reviewer #1: In this paper, the authors set out to explore gender parity in the context of gender representation and internal collaboration at the Cyprus Institute.

The heart of this paper is a nice experimental design with results that will be of interest to at least the STEM community, but I have some concerns about the paper in its current forms:

• The literature should discuss relevant studies about gender-gap to a sufficient depth and the structure of the arguments.

• How is gender in Cyprus's cultural context different/the same as how gender works in the cultural contexts of previously published studies?

• Why do we need this study and rationale? What hole does it fill?

• There has been a long-running conversation about gender gaps, and the authors need to establish where their research falls in that conversation.

The literature cited is not sufficient to make the following statements

"Whilst it does qualify for inclusion in the comparative gender analysis, that in itself indicating a positive development in terms of numbers of academics publishing, it demonstrates a statistically significant gender gap with respect to annual and total productivity, total impact, and career length with better performance by male academic.

• The axis of the graphs is not clear.

• I'd like to see a table in your methods of each section in the study and its composition. Right now, it's a little challenging to understand your experimental setup.

• I find the manuscript suffers from a weak argument about its implications and this should be very clear.

Reviewer #2: The present work, show a very interesting analysis about the status of gender equality in The Cyprus Institute of Neurology and Genetics of Cyprus (CING). In general, the information showed is clear and the statical analysis support the authors observation.

My only observation is that, the information showed, just represent the characteristics of the CING. The next step could be a deeper study that show how the gender equality status is observed in other institutions of Cyprus. Likewise, in my opinion, this analysis show a very important advances in the equality state between woman and man and, it is the results of many changes in the organization of the CING. Might be, the description of how this institution obtaind these results should be mentioned in the discussion to help others to improve their regulations.

6. PLOS authors have the option to publish the peer review history of their article (what does this mean?). If published, this will include your full peer review and any attached files.

Reviewer #1: **Yes: **Firas Almasri

Reviewer #2: **Yes: **Arturo Aguilar-Rojas

---

## [Author Response · Author response to Decision Letter 0]

18 Jul 2022

• Reviewers' comments:

Reviewer's Responses to Questions

Comments to the Author

1. Is the manuscript technically sound, and do the data support the conclusions?

Reviewer #1: Partly

Reviewer #2: Yes

We have now added two additional tables in the Methods section to clarify the technical Methods used in each section.

2. Has the statistical analysis been performed appropriately and rigorously?

Reviewer #1: Yes

Reviewer #2: Yes

3. Have the authors made all data underlying the findings in their manuscript fully available?

Reviewer #1: Yes

Reviewer #2: Yes

4. Is the manuscript presented in an intelligible fashion and written in standard English?

Reviewer #1: No

Reviewer #2: Yes

We have now reviewed the manuscript and have corrected any typographical or grammatical errors.

5. Review Comments to the Author

Reviewer #1: In this paper, the authors set out to explore gender parity in the context of gender representation and internal collaboration at the Cyprus Institute.

The heart of this paper is a nice experimental design with results that will be of interest to at least the STEM community, but I have some concerns about the paper in its current forms:

• The literature should discuss relevant studies about gender-gap to a sufficient depth and the structure of the arguments.

We have now added additional relevant studies with respect to the gender gap to better structure and enforce our arguments with respect to that in the Introduction section. Specifically, we have included relevant studies and discussion of gender gap in relation to power and leadership, grant awards and science prizes. We have also discussed the underlying causes of gender gap in a wider global context and the reasons why closing the gender gap is considered important. Introduction: Lines 122-137, 179-211 and discussion: 739-747. 

• How is gender in Cyprus's cultural context different/the same as how gender works in the cultural contexts of previously published studies?

We have included a discussion on the cultural context with respect to gender inequality in Cyprus compared to other countries and in addition have presented from a historical perspective the state and improvements observed in the recent years as well the areas in need of further improvement: Lines 76-120. We have added a point regarding our findings with respect to the cultural context in Cyprus in the Discussion: Lines 815-818.

• Why do we need this study and rationale? What hole does it fill?

We have now clarified the rational of this study and the hole that it fills in the part of the Introduction: Lines 255-262

• There has been a long-running conversation about gender gaps, and the authors need to establish where their research falls in that conversation. The literature cited is not sufficient to make the following statements

"Whilst it does qualify for inclusion in the comparative gender analysis, that in itself indicating a positive development in terms of numbers of academics publishing, it demonstrates a statistically significant gender gap with respect to annual and total productivity, total impact, and career length with better performance by male academic.

We have deleted the statement “Whilst it does qualify for inclusion in the comparative gender analysis, that in itself indicating a positive development in terms of numbers of academics publishing” and clarified that the rest of the statement“, it demonstrates a statistically significant gender gap with respect to annual and total productivity, total impact, and career length with better performance by male academic” pertains to the Huang et al., PNAS. 2020. In addition, we have elaborated further on the data presented in this study and the comparative position of Cyprus within that: Lines 162-177.

• The axis of the graphs is not clear.

The axes have been edited to increase clarity of the axes in Figures 1-4. No further action was deemed necessary for the rest of the graphs.

• I'd like to see a table in your methods of each section in the study and its composition. Right now, it's a little challenging to understand your experimental setup.

We have now added two tables in the Methods section summarizing the methods performed in each of the two major sections (1. Gender Distribution of Employees in the CING and 2. Collaborativeness in CING Co-authorship Network). Note that the first table is a modified extended version of the original Table 1 with analysis categories which was previously included in the manuscript.

• I find the manuscript suffers from a weak argument about its implications and this should be very clear.

We have now clarified in the Discussion (Conclusions) the implications of our work: Lines: 796-818, 847-853 

Reviewer #2: The present work, show a very interesting analysis about the status of gender equality in The Cyprus Institute of Neurology and Genetics of Cyprus (CING). In general, the information showed is clear and the statical analysis support the authors observation.

My only observation is that, the information showed, just represent the characteristics of the CING. The next step could be a deeper study that show how the gender equality status is observed in other institutions of Cyprus. Likewise, in my opinion, this analysis shows a very important advances in the equality state between woman and man and, it is the results of many changes in the organization of the CING. Might be, the description of how this institution obtained these results should be mentioned in the discussion to help others to improve their regulations.

We have now added a part in the Discussion comparing our work with disaggregated data published in a preliminary report and policy document of one comparable local organization’s results (Lines 777-791). However, as we also state in the manuscript, an accurate comparative analysis with the data of that report cannot be performed at this stage since these data were not stratified in accordance with the subspecialties within the biological sciences as we have analysed our work.

We also now mention in the Discussion that the combination of internal policies in place for equal opportunities as well as state accreditation and EU monitoring have contributed to the attainment of these results: Lines:796-818.

---

## [Editor Report · Decision Letter 1]

26 Aug 2022

The Cyprus Institute of Neurology and Genetics, an Emerging Paradigm of a Gender Egalitarian Organisation

PONE-D-21-38904R1

Dear Dr. Xenophontos,

We’re pleased to inform you that your manuscript has been judged scientifically suitable for publication and will be formally accepted for publication once it meets all outstanding technical requirements.

Kind regards,

Dr Aleksandra Barac

Academic Editor

PLOS ONE

---

## [Editor Report · Acceptance letter]

2 Sep 2022

PONE-D-21-38904R1 

The Cyprus Institute of Neurology and Genetics, an Emerging Paradigm of a Gender Egalitarian Organisation 

Dear Dr. Xenophontos:

I'm pleased to inform you that your manuscript has been deemed suitable for publication in PLOS ONE. Congratulations! Your manuscript is now with our production department. 

Kind regards, 

on behalf of

Dr. Aleksandra Barac 

Academic Editor

PLOS ONE